# Predicting Adolescent Internet Gaming Addiction from Perceived Discrimination, Deviant Peer Affiliation and Maladaptive Cognitions in the Chinese Population: A Two-Year Longitudinal Study

**DOI:** 10.3390/ijerph19063505

**Published:** 2022-03-16

**Authors:** Likun Wang, Meijin Li, Yang Xu, Chengfu Yu

**Affiliations:** 1School of Data Science and Computer Science, Sun Yat-Sen University, 510006 Guangzhou, China; wanglikun2022@163.com; 2Organization Department, The Chinese Communist Party Committee of Guangdong Province, 510080 Guangzhou, China; 3Research Center of Adolescent Psychology and Behavior, Department of Psychology, School of Education, Guangzhou University, 510006 Guangzhou, China; 2112008030@e.gzhu.edu.cn; 4School of Psychology, South China Normal University, 510631 Guangzhou, China; xuyang0418@yeah.net

**Keywords:** adolescents, perceived discrimination, Internet gaming addiction (IGA), deviant peer affiliation, maladaptive cognitions

## Abstract

A robust positive association between perceived discrimination and Internet gaming addiction (IGA) among adolescents has been demonstrated by existing research; however, the mediating mechanisms underlying this relationship remain largely unknown. This study, grounded in the cognitive–behavioral and social development models, examined whether deviant peer affiliation and maladaptive cognitions mediated the predictive effect of perceived discrimination on adolescent IGA. Six-hundred-and-sixty students (Mean_age_ = 13.43 years; 367 female participants) recruited from southern China participated in four assessments (fall 7th grade, spring 7th grade, fall 8th grade, spring 8th grade). Participants completed a self-administered questionnaire that assessed their demographics, as well as questionnaire measures of perceived discrimination, deviant peer affiliation, maladaptive cognitions, and IGA. The structural equation model showed that fall 7th grade’s perceived discrimination positively predicted spring 7th grade’s deviant peer affiliation, which, in turn, increased fall 8th grade’s maladaptive cognitions, ultimately increasing spring 8th grade’s IGA. Moreover, the indirect effect of fall 7th grade’s perceived discrimination on spring 8th grade’s IGA via spring 7th grade’s deviant peer affiliation was also significant. The results suggested deviant peer affiliation and maladaptive cognitions as potential mediating mechanisms linking perceived discrimination to adolescent IGA. These findings have important implications for the prevention and intervention of adolescent IGA.

## 1. Introduction

Internet gaming addiction (IGA) is a subtype of Internet addiction that refers to the uncontrollable, excessive, and compulsive use of online games that causes social and/or emotional problems [1]. In June 2021, the China Internet Network Information Center survey showed that the number of young Internet users aged 10–19 reached 158 million, accounting for 12.3% of all Internet users. Survey research has revealed that Chinese adolescents have higher IGA prevalence rates (14.6%) than their peers in most other countries, including Indonesia, Slovenia, and Singapore [2,3,4]. Furthermore, empirical studies have indicated that IGA could lead to adolescent major depressive disorder, social anxiety disorder, suicidal thoughts, and suicide attempts [5]. Therefore, it is crucial to understand the mechanisms underlying IGA among adolescents for developing targeted prevention programs. This study examined the predictive effects of perceived discrimination, deviant peer affiliation, and maladaptive cognitions on adolescent IGA to address this issue through a two-year longitudinal study.

### 1.1. Perceived Discrimination and Adolescent IGA

Perceived discrimination is a prominent risk factor for adolescent development [6]. It refers to an individual’s perception that they are unfairly treated due to their group membership, such as race, gender, socioeconomic status, and so on. This bias can be demonstrated by actual actions and rejecting attitudes [7]. Numerous empirical studies have shown that adolescents’ perceived discrimination could significantly predict internalizing problems (e.g., depression) and various addictive behaviors (e.g., substance use and IGA) [8,9]. As suggested by the stress–coping model [10], addictive behaviors such as IGA often serve as a means to cope with the adverse effects of perceived discrimination. In addition, frequent experiences with discrimination may increase negative emotions in adolescents who lack psychological maturity and effective coping strategies and, thus, may turn to online games to escape from the stress and dissatisfaction of real life. Research has also indicated that adolescents who perceive discrimination are more likely to develop IGA [11,12]. These findings highlight the essential contribution of perceived discrimination in increasing adolescent IGA. However, previous studies have primarily focused on the direct link between perceived discrimination and adolescent IGA. In contrast, the mediating mechanism (how does perceived discrimination lead to IGA?) remains largely unknown. In this study, which is rooted in the cognitive–behavioral [13] and social development models [14], we aimed to test whether deviant peer affiliation and maladaptive cognitions mediated the relationship between perceived discrimination and adolescent IGA.

### 1.2. Deviant Peer Affiliation as a Mediator

Deviant peer affiliation refers to an abnormal connection with a peer group, in which members share similar external or internal problems and influence each other in negative ways [15]. The influence of peers on adolescents generally increases throughout adolescence. According to the social development model [14], adolescents develop positive social bonding if they have positive experiences interacting with others and are involved in activities that can prevent adolescent affiliation with deviant peers, inhibiting problem behavior. Conversely, perceived discrimination, as a negative interpersonal interaction, may impair adolescents’ attachment to others and their belief in compliance with social norms, thus increasing the possibility of deviant peer affiliation and further development of IGA. In other words, deviant peer affiliation may mediate the relationship between perceived discrimination and IGA. Several studies have indicated that deviant peer affiliation mediates negative interpersonal interactions and problem behaviors [16,17,18].

On the one hand, adolescents who perceive discrimination or unfair treatment from the mainstream peer group may seek emotional support from other alternative groups, such as deviant youths who are also rejected by the mainstream peer group [16]. For example, Brody et al. [17] found that perceived discrimination predicts greater affiliations with substance-using peers and, in turn, greater substance use. On the other hand, adolescents may develop IGA through peer pressure and imitation [19,20]. For instance, research has shown that adolescents who associate with friends addicted to computer games have a higher risk of IGA because deviant friends may provide more opportunities for them to play online games [21].

Based on the literature reviewed above, we propose the following hypothesis:

**Hypothesis** **1** **(H1).***Deviant peer affiliation mediates the relationship between perceived discrimination and IGA among adolescents*.

### 1.3. Maladaptive Cognitions as a Mediator

Maladaptive cognitions refer to the cognitive bias that individuals form toward themselves and the world after using the Internet [13]. Based on Davis’ cognitive–behavioral model, IGA-related maladaptive cognitions are the critical factor for the onset and development of IGA. Using a 12-month longitudinal study, Forrest et al. [22] found that IGA-related maladaptive cognitions predicted changes in IGA. Another longitudinal study confirmed that maladaptive cognitions and emotions related to users’ problematic online gaming represent a powerful mediating mechanism for maintaining this behavior [23]. Further, Liu et al. [24] reported that maladaptive cognitions mediate adolescents’ psychological distresses and problematic mobile phone use. Research has recently shown that maladaptive cognition plays a pivotal mediating role in perceived relationship conflict and problematic social media use [25]. Study also indicates that adolescents’ discrimination experiences are associated with poor academic achievement, low self-esteem, and low satisfaction of psychological needs [26,27]. This relationship may, in turn, make it easier for adolescents to develop maladaptive cognitions once they start playing online games. Based on the cognitive–behavioral model and previous relevant studies, it can be inferred that IGA results from problematic cognitions and behaviors. Thus, adolescents who suffer from discrimination may be more likely to develop maladaptive cognitions than those who do not suffer from it which, in turn, increases the likelihood of IGA.

According to this literature, we propose our second hypothesis:

**Hypothesis** **2** **(H2).***Maladaptive cognitions will mediate the relationship between perceived discrimination and IGA among adolescents*.

### 1.4. A Multiple Mediation Model

Although deviant peer affiliation or maladaptive cognitions may play an independent mediating role in the relationship between perceived discrimination and IGA, other mediation models may more clearly explain the effects of perceived discrimination on IGA [28]. The increased risk of maladaptive cognitions may be directly affected by perceived discrimination or indirectly affected by deviant peer affiliation, thus developing IGA. First, due to perceived discrimination or unfair treatment from group members, adolescents are highly inclined to affiliate with deviant peers who have similar experiences, opinions, and problem behaviors as their best friends [16,17]. Second, in the case of adolescents affiliated with deviant peers who have similar views, external or internal problems in the group are more likely to develop maladaptive cognition [29,30]. Third, individuals who experience negative peer relationships view things more negatively, resulting in maladaptive cognition [31]. Finally, maladaptive cognitions may readily induce adolescents’ extreme thoughts that the real self is worthless and the online self is valuable, thus developing greater IGA [13,32].

Based on the empirical research referred to above, we propose the following hypothesis:

**Hypothesis** **3** **(H3).***The indirect effect of perceived discrimination on IGA is mediated by deviant peer affiliation and maladaptive cognitions*.

### 1.5. The Present Study

Based on the cognitive–behavioral [13] and social development models [14], we used a two-year longitudinal design to explain how perceived discrimination predicts IGA and emphasized the mediating role of deviant peer affiliation and maladaptive cognitions in this process. Specifically, we aimed to examine (1) whether spring 7 grade’s deviant peer affiliation mediated the relationship between fall 7 grade’s perceived discrimination and spring 8 grade’s IGA; (2) whether fall 8 grade’s maladaptive cognitions acted as a mediator in the link between fall 7 grade’s perceived discrimination and spring 8 grade’s IGA; and (3) whether spring 7 grade’s deviant peer affiliation and fall 8 grade’s maladaptive cognitions sequentially mediated the above pathway. Figure 1 illustrates the proposed research model.

## 2. Materials and Methods

### 2.1. Participants

Participants were recruited from three junior middle schools in Guangdong Province, a southern province in China, through stratified and random cluster sampling. Data collection was performed four times at six-month intervals. At Time 1 (fall of 7th grade), 725 adolescents participated in the study; at Time 4 (spring of 8th grade), 660 adolescents completed the follow-up measurement. The mean age of the 660 final participants was 13.43 years (SD = 0.48, 367 female participants) at Time 1. There were non-significant differences between the included and excluded adolescents in any of the variables.

### 2.2. Procedure

We provided students interested in participating in the survey with consent forms for their parents/guardians to sign several days prior to the administration of the survey. They were assured that the survey was anonymous and confidential and that they could withdraw at any time without penalty. With the assistance of the school’s head of guidance, the second author and research assistants administered the surveys at the school during a 30 min session. In addition, data collectors were available to respond to the participants’ questions throughout survey administration. No financial incentive was given to the participants for completing the survey. The survey materials and study procedures were approved by the Ethics in Human Research Committee of the Department of Psychology, Guangzhou University (protocol number: GZHU2019012; date of approval: 27 May 2019).

### 2.3. Measures

#### 2.3.1. Perceived Discrimination

In line with previous studies [17], at Time 1, participants responded to 10 items about perceived discrimination during the past six months (e.g., “I was considered dishonest because I am an outsider or my parents are not around”) on a 3-point scale: 1 (never) to 3 (frequently). Item scores were averaged to create a composite of perceived discrimination, with higher scores indicating higher levels of perceived discrimination. The Cronbach’s alpha value was 0.92.

#### 2.3.2. Deviant Peer Affiliation

At Time 2, deviant peer affiliation was measured using seven items adopted from previously published questionnaires [33,34]. Respondents were asked to indicate how many of their close friends displayed behaviors such as “fighting and brawling” in the last six months. All items were rated on a 5-point scale ranging from 1 (none) to 5 (six or more). Item scores were averaged to create a composite of deviant peer affiliation. Higher scores indicated higher levels of deviant peer affiliation. The Cronbach’s alpha value was 0.77.

#### 2.3.3. Maladaptive Cognitions

Maladaptive cognitions were measured at Time 3 with a 12-item Chinese version of the Maladaptive Cognitions Questionnaire [32]. Respondents were asked to report their thoughts about the Internet during the past six months (e.g., “Online friends will tell you more true words”). All items were rated on a 5-point scale ranging from 1 (strongly disagree) to 5 (strongly agree). Item scores were averaged to create a composite of maladaptive cognitions, with higher scores indicating higher levels of maladaptive cognitions. The Cronbach’s alpha value was 0.93.

#### 2.3.4. IGA

IGA was measured at Time 4 with an 11-item Chinese version of the Problematic Online Game Use Questionnaire [35]. Respondents were asked about playing computer games during the past six months (e.g., “Do you feel the need to use the Internet game for increasing amounts of time to achieve satisfaction”). All items were rated on a 3-point scale ranging from 1 (never) to 3 (frequently). Item scores were averaged to create a composite of IGA, with higher scores indicating higher levels of IGA. The Cronbach’s alpha value was 0.90.

#### 2.3.5. Control Variables

Given that adolescent gender, age, and self-esteem were associated with IGA [35], the statistical models included these factors as control variables. Self-esteem was assessed by the Rosenberg Self-Esteem Scale [36]. The mean of all items was calculated, with higher scores reflecting higher levels of self-esteem. The Cronbach’s alpha value was 0.85.

### 2.4. Statistical Analysis

All statistical analyses were conducted using Mplus version 7.4. (Tihomir Asparouhov and Thuy Nguyen Team, Los Angeles, CA, USA). First, we performed descriptive analysis and correlation analysis to determine the mean and standard deviation and the correlations between the variables in this study. Moreover, we tested mediation effects using structural equation modeling with maximum likelihood estimation and bootstrapping with 1000 replications in Mplus 7.4.

## 3. Results

### 3.1. Preliminary Analyses

Means, standard deviations, and correlations among the study variables are shown in Table 1. As we hypothesized, the correlations among the four variables were significant. Perceived discrimination, deviant peer affiliation, and maladaptive cognitions were positively correlated with IGA. In addition, perceived discrimination and deviant peer affiliation were positively correlated with maladaptive cognitions. Finally, perceived discrimination was positively correlated with deviant peer affiliation.

### 3.2. Testing for Mediation Effects

We tested the multiple mediation model as follows: First, we evaluated the fit of the full-proposed model (Model 1). This model is a saturation model. Consistent with the hypothesized paths, fall 7th grade’s perceived discrimination was associated with higher spring 7th grade’s deviant peer affiliation, which in turn predicted higher fall 8th grade’s maladaptive cognitions, which ultimately predicted spring 8th grade’s IGA. Moreover, the indirect effect of “fall 7th grade’s perceived discrimination → spring 7th grade’s deviant peer affiliation → spring 8th grade’s IGA” was also significant. However, the direct effects of “fall 7th grade’s perceived discrimination → fall 8th grade’s maladaptive cognitions,” and “fall 7th grade’s perceived discrimination → spring 8th grade’s IGA” were not significant. The mediation effect inspection path graph is shown in Figure 2.

Then, we evaluated the fit of the reduced model (Model 2), in which the non-significant paths were dropped from the previous model. The results showed that the reduced model’s fit was excellent: *χ*^2^/*df* = 4.20, CFI = 0.978, TLI = 0.836, RMSEA = 0.070. The mediation effect inspection path graph is shown in Figure 3. Compared with the full model, the reduced model did not have a significantly worse fit, so we selected the more compact one. Thus, Model 2 was the final model. In this model, the indirect effect “perceived discrimination → deviant peer affiliation → maladaptive cognitions → IGA” was significant (indirect effects = 0.02, 95% CI (0.005, 0.025]). Moreover, the indirect effect “perceived discrimination → deviant peer affiliation → IGA” was also significant (indirect effect = 0.05, 95% CI [0.010, 0.060]).

## 4. Discussion

Based on integrating the cognitive–behavioral model [13] and the social development model [14], this prospective study utilized a two-year longitudinal design to identify predictors of adolescent IGA. As hypothesized, perceived discrimination was related to adolescent IGA via the sequential mediation of deviant peer affiliation and maladaptive cognitions. This is the first study to examine the relationship between these variables.

### 4.1. The Mediating Effect of Deviant Peer Affiliation and Maladaptive Cognitions

Consistent with Hypothesis 1, we found that deviant peer affiliation was an important underlying psychosocial mechanism that helped to explain why perceived discrimination was associated with IGA. According to the social development model [14], when adolescents are discriminated against, they are more likely to affiliate with deviant peers related to IGA. Adolescents who experience perceived discrimination may have negative attitudes toward mainstream peer groups and may avoid associating with them [37,38]. When they are distant from mainstream peer groups or perceive peer discrimination, adolescents are likely to associate with deviant peers who have similar views and experiences by homophily selection [29,39]. As a result, through the influence of peer norms, modeling, and pressure, adolescents are more likely to increase the risk of IGA. For example, Brody et al. [17] demonstrated that perceived discrimination was a powerful predictor of deviant peer affiliation and, in turn, caused addiction behaviors among adolescents. However, the findings of the current study did not support Hypothesis 2. Thus, the indirect effect of perceived discrimination on IGA via maladaptive cognitions was insignificant. A possible reason for this is that the maladaptive cognitions related to online games are due to affiliations with peers who also have IGA rather than being directly influenced by perceived discrimination.

### 4.2. The Multiple Mediation Model of Deviant Peer Affiliation and Maladaptive Cognitions

Consistent with Hypothesis 3, the findings supported the predicted multiple mediation model. Specifically, fall 7th grade’s perceived discrimination was associated with increased spring 7th grade’s deviant peer affiliation, which in turn increased fall 8th grade’s maladaptive cognitions and, ultimately, spring 8th grade’s IGA. According to Davis’ cognitive–behavioral model [13], maladaptive cognitions are the key factor in the onset and development of IGA [23,32]. Meanwhile, deviant peer affiliation is closely related to maladaptive cognitions [31]. However, most prior empirical research has focused on the influence of these two factors (maladaptive cognitions and deviant peer affiliation) separately.

One strength of the current study was that we included deviant peer affiliation and maladaptive cognitions in the same “multiple mediation” model of IGA. Earlier research showed that deviant peer affiliation and maladaptive cognitions each play important mediating roles in the association between perceived discrimination and adolescent IGA. The current results are consistent with earlier research, showing that adolescents who affiliate with deviant peers would promote risky behaviors [40]. Specifically, those who perceive discrimination usually have related emotional problems [41]; to find emotional support, they may associate with deviant peers [42,43]. Furthermore, the resulting social exclusion may foster maladaptive cognitions [30], which is the crucial factor for the onset and development of IGA [32]. The current results suggest that weakening adolescents’ deviant peer affiliations and maladaptive cognitions may decrease the risk of IGA.

In the current study, perceived discrimination appeared to have a cascade effect leading to deviant peer affiliation and contributing to subsequent maladaptive cognitions and further development of IGA. One of the critical advances of our model is the focus on deviant peer affiliation and maladaptive cognitions as important mediators; the support of this sequential mediation model makes essential contributions to future studies.

### 4.3. Implications

The present study has important practical implications. First, our results suggest that perceived discrimination is an important factor leading to IGA, offering a possible focus for prevention. Second, our findings can help practitioners understand the pathways through which perceived discrimination is associated with adolescent IGA, suggesting a potential direction for targeted interventions. For example, reducing deviant peer affiliation may alleviate some of the detrimental effects of perceived discrimination on adolescent IGA. This effect is crucial because many of the current IGA interventions are delivered in peer-group settings (e.g., adolescents with IGA were aggregated into groups such as training camps), which, under some circumstances, may inadvertently increase adolescent interaction with deviant peers [44]. Moreover, the results suggest that reducing maladaptive cognitions might ameliorate some detrimental effects of perceived discrimination on adolescent IGA.

## 5. Conclusions

This study is an important step in investigating the relationship between perceived discrimination and adolescent IGA. We found that perceived discrimination was positively associated with IGA, while deviant peer affiliation can potentially mediate between perceived discrimination and IGA. Moreover, our findings suggest that maladaptive cognitions act as a mediator by which deviant peer affiliation predicts IGA. In other words, the results suggested that deviant peer affiliation and maladaptive cognitions as potential mediating mechanisms linking perceived discrimination to adolescent IGA. These findings, taken together, strengthen the current scholarly understanding of the potential mediating mechanisms of perceived discrimination on IGA, which brings clear implications for possible interventions.

However, several limitations must be considered when interpreting the results of this study. First, previous research has shown that adolescent self-reports of risk behaviors are not strongly biased under conditions of anonymity and confidentiality [45,46]. Still, the method of self-report may lead to memory bias, as was the case in our study. Future studies should simultaneously employ multiple informants and multiple data collection methods to obtain more objective evidence. Second, the model was tested using a sample of Chinese adolescents. Therefore, we should not generalize the current conclusions to other cultural and geographical settings.

## Figures and Tables

**Figure 1 ijerph-19-03505-f001:**
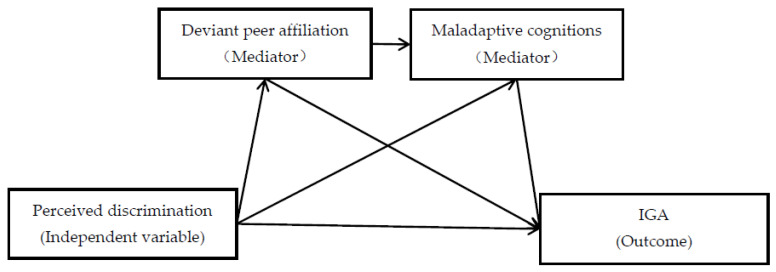
The proposed mediation model. The path “perceived discrimination → deviant peer affiliation → adolescent IGA” tests Hypothesis 1, the path “perceived discrimination → maladaptive cognitions → adolescent IGA” tests Hypothesis 2, and the path “perceived discrimination → deviant peer affiliation → maladaptive cognitions → adolescent IGA” tests Hypothesis 3.

**Figure 2 ijerph-19-03505-f002:**
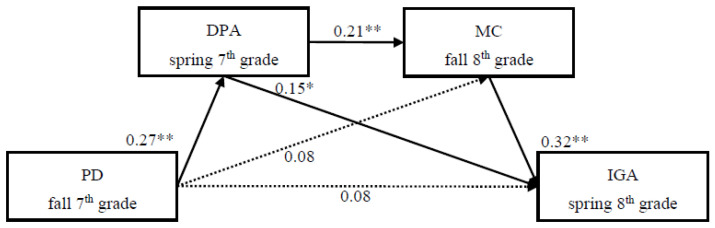
Full model (Model 1) for the relationship between peer victimization (PD) and Internet gaming addiction (IGA) was mediated by deviant peer affiliation (DPA) and maladaptive cognitions (MC). This model is a saturation model. Significant standardized paths are displayed with solid lines; no significance is shown with dotted lines. * *p* < 0.05, ** *p* < 0.01.

**Figure 3 ijerph-19-03505-f003:**
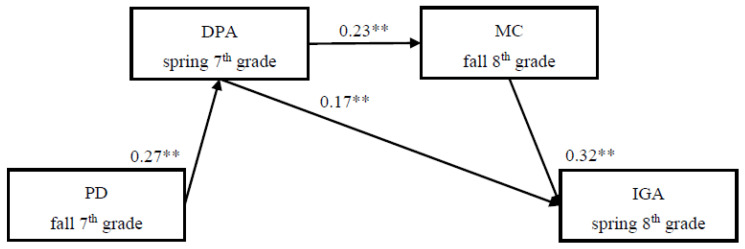
Reduced model (Model 2) for relationship between peer victimization (PD) and Internet gaming addiction (IGA) were mediated by deviant peer affiliation (DPA) and maladaptive cognitions (MC). *χ*^2^/*df* = 4.20, CFI = 0.978, TLI = 0.836, RMSEA = 0.070. ** *p* < 0.01.

**Table 1 ijerph-19-03505-t001:** Descriptive statistics and bivariate correlations between all study variables.

Variable	1	2	3	4	5	6	7
1. Gender	1.00						
2. Age	0.03	—					
3. Self-esteem	−0.09 *	−0.05	—				
4. PD (T1)	0.15 **	0.03	−0.32 **	—			
5. DPA (T2)	0.06	0.07	−0.09 **	0.28 **	—		
6. MC (T3)	0.11 **	0.13 **	−0.17 **	0.18 **	0.25 **	—	
7. IGA (T4)	0.26 **	0.04	−0.13 **	0.22 **	0.27 **	0.39 **	—
Mean	0.56	13.43	2.94	1.27	1.20	1.90	1.16
SD	0.50	0.48	0.54	0.38	0.36	0.76	0.25

N = 660. Gender was dummy coded such that 0 = male and 1 = female. PD = perceived discrimination, DPA = deviant peer affiliation, MC = maladaptive cognitions, IGA = Internet gaming addiction. T1 *=* fall 7th grade; T2 *=* spring 7th grade; T3 *=* fall 8th grade; T4 *=* spring 8th grade. * *p* < 0.05. ** *p* < 0.01.

## Data Availability

The data presented in this study are available on request from the corresponding authors (C.Y.).

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
