# Peer review of "Predicting Adolescent Internet Gaming Addiction from Perceived Discrimination, Deviant Peer Affiliation and Maladaptive Cognitions in the Chinese Population: A Two-Year Longitudinal Study"

_ijerph, 2022, doi:10.3390/ijerph19063505_

Round 1

Reviewer 1 Report

Dear authors, please look at the questions presented in this review, some of which are of profound interest.

This paper presents a potentially goal, which is to examine whether deviant peer affiliation and maladaptive cognitions mediated the predictive effect of perceived discrimination on adolescent IGA. Unfortunately, there are very relevant content issue that severely limit the validity of the study. The first question relates to the paper itself, it constitutes an original contribution doing a two-years longitudinal study.

I suggest modifying the title, adding “to the Chinese population” whether it can be considered by editors. Furthermore, the authors should explain the choice for 7th and 8th grades and not using 6th grade also, given that IGA starts in early years.

There is another methological doubt, and I would like the authors to explain to me, the linear relationship of the grades is: 7th grade in spring 7th grade in fall, 8th in spring and 8th in fall or the two grades participate in parallel mode and the results are compared.

I have my doubts about the extent to which recall can be captured from participants in the variables used and if it is a good method of getting accurate and precise information. It would affect the results if the memories experienced are not precisely remembered.

“First, we evaluated the fit of the full proposed model (Model 1). After controlling for variables representing gender, age, and self-esteem, the results showed that the full model fit was excellent: 2/df =0.000, CFI=1.000, TLI=1.000, RMSEA=0.000”.

Achieving a perfect model is highly impossible because it would be understood that the error factor, produced either in the collection of data or in the treatment of the variables, would not exist at random and it is statistically impossible. Because all the variance would be due to the effect of the variable excluding the error.

I must say, I consider the use of both cognitive-behavioral and social developmental theoretical models to be appropriate. One question needs to be explained, Table 1 shows variable number 3 called self-esteem, but throughout the text it does not appear how the results were obtained, whether it is a single question or whether a scale was used. The authors should clarify this question in the text.

As well, I found the conclusion too brief and the limitations listed in the discussion section should be changed to conclusions section.

Author Response

Comment 1: I suggest modifying the title, adding “to the Chinese population” whether it can be considered by editors. Furthermore, the authors should explain the choice for 7th and 8th grades and not using 6th grade also, given that IGA starts in early years.

Response: Thank you for your suggestion. We have add “in the Chinese population” in the title. Moreover, The selection of subjects is related to the division of Chinese education stages. 6th grade students still belong to the primary school stage. However, students from Grade 7 to grade 8 are belonging to middle school stage. During middle school stage, students will pay more attention to their inner feelings and the evaluation of others, so they are more likely to perceive discrimination from peers and society, which leads to the affiliation with deviant peers and ultimately promotes IGA.

Comment 2:There is another methological doubt, and I would like the authors to explain to me, the linear relationship of the grades is: 7th grade in spring 7th grade in fall, 8th in spring and 8th in fall or the two grades participate in parallel mode and the results are compared.

Response: We are extremely grateful to reviewer for pointing out this problem. The first assessment (Time 1) was conducted in the autumn semester of Grade 7 on October 2019 (7th grade in fall), and then it was conducted every six months. For example, the assessments of 7th grade in spring (Time 2) and 8th in spring (Time 3) was seperately completed on April 2020 and October 2020. A total of 4 tests were conducted, and the students were tracked until the spring of Grade 8 (Time 4).

Comment 3:I have my doubts about the extent to which recall can be captured from participants in the variables used and if it is a good method of getting accurate and precise information. It would affect the results if the memories experienced are not precisely remembered.

Response: Thanks for your valuable suggestion. It is true that the method of adolescent self-report will lead to some memory bias. We have listed the methological limitations of this method in the conclusion section. Future studies should simultaneously employ multiple informants and multiple data collection methods.

Modification in text as follows:

  However, several limitations must be considered when interpreting the results of this study. First, although previous research has shown that adolescent self-reports of risk behaviors are not strongly biased under conditions of anonymity and confidentiality [42,43] and the method of self-report may lead to memory bias, as was the case in our study, future studies should simultaneously employ multiple informants, multiple data collection methods and find more objective evidence.

Comment 4:“First, we evaluated the fit of the full proposed model (Model 1). After controlling for variables representing gender, age, and self-esteem, the results showed that the full model fit was excellent: c2/df =0.000, CFI=1.000, TLI=1.000, RMSEA=0.000”. Achieving a perfect model is highly impossible because it would be understood that the error factor, produced either in the collection of data or in the treatment of the variables, would not exist at random and it is statistically impossible. Because all the variance would be due to the effect of the variable excluding the error.

Response: Thank you for your suggestion. Model 1 is a saturation model. The fitting index of saturation model doesn't make sense. It is worth noting that we also present the final model (Model 2). The fitting index of the model 2 was excellent.

Comment 5:I must say, I consider the use of both cognitive-behavioral and social developmental theoretical models to be appropriate. One question needs to be explained, Table 1 shows variable number 3 called self-esteem, but throughout the text it does not appear how the results were obtained, whether it is a single question or whether a scale was used. The authors should clarify this question in the text.

Response: Thank you for your suggestion. Given that adolescent’s self-esteem was associated with IGA [35], self-esteem was included as control variables in the statistical models. Self-esteem was assessed by the Rosenberg Self-Esteem Scale [36]. The mean of all items was calculated, with higher scores reflecting higher levels of self-esteem. The Cronbach’s alpha was 0.85.

Comment 6:As well, I found the conclusion too brief and the limitations listed in the discussion section should be changed to conclusions section.

Response: Thank you for your suggestion. According to the reviewer's suggestions, we supplement the conclusion part and the limitations listed in the discussion section had been changed to conclusions section.

Reviewer 2 Report

The manuscript presents an interesting and applicable study for the improvement of adolescent mental health. The following aspects should be reviewed:

The summary does not make clear the conclusions of the study.

More current bibliographical references should be included.

The number of missing subjects, excluded from the final sample because they did not complete all the questionnaires over the four data collections, should be included in the material and methods section.

The statistical tests carried out should also be described in the methodology section.

Author Response

Comment 1:The summary does not make clear the conclusions of the study.

Response: Thank you for your suggestion. According to the reviewer's suggestions,we have supplemented the conclusion section.

Comment 2:More current bibliographical references should be included.

Response: Thank you for your suggestion. According to the reviewer's suggestions,we have added bibliographical references as follows:

The current results are consistent with earlier research which showed that adolescents who affiliate with deviant peers would promote risky behaviors[40], specifically, perceive discrimination usually have corresponding emotional problems [41]; to find emotional support, they may associate with deviant peers [42,43]. line 287-290.

[40] Jessor, R.; Donovan, J.E.; Costa, F.M. Beyond adolescence: Problem behavior and young adult development. New York, NY: Cambridge University Press. 1991.

[43] Parker, J.G.; Rubin, K.H; Erath, S.A.; Wojslawowicz, J.C.; Buskirk, A.A. Peer relationships, child development, and adjustment: a developmental psychopathology perspective. In: Cicchetti D, Cohen DJ (eds) Developmental psychopathology, 2nd edn. Wiley, Hoboken, 2006, pp 419–493

Comment 3:The number of missing subjects, excluded from the final sample because they did not complete all the questionnaires over the four data collections, should be included in the material and methods section.

Response: Thank you for your suggestion. We have add the information of the missing subjects in the material and methods section.

Comment 4: The statistical tests carried out should also be described in the methodology section.

Response: Thanks for your valuable suggestion. We have add the statistical analysis in the methodology section.

Round 2

Reviewer 2 Report

The authors have responded to the reviewers' suggestions by improving the manuscript.
In the abstract of the submitted manuscript, the conclusions of the study should be included, as they now appear in an unclear form. This aspect should be improved.

Author Response

Thanks again the editor and the anonymous reviewers for the insightful comments on our manuscript. This study has been revised according to the comments of the reviewer.

Firstly, In the abstract of the manuscript, the conclusions of the study has been supplemented.

Secondly, after reading through the full text, all the authors proofread the full text again and then polished the language expression of the article.